# Unrecognized and Unreported Concussions Among Community Rugby Players

**DOI:** 10.3390/sports13080278

**Published:** 2025-08-20

**Authors:** Rachael Wittmer, Thomas A. Buckley, Charles Buz Swanik, Katelyn M. Costantini, Lisa Ryan, Ed Daly, Regan E. King, Arryana J. Daniels, Katherine J. Hunzinger

**Affiliations:** 1Sidney Kimmel Medical College, Thomas Jefferson University, Philadelphia, PA 19107, USA; rachael.wittmer@jefferson.edu (R.W.); regan.king@students.jefferson.edu (R.E.K.); arryana.daniels@students.jefferson.edu (A.J.D.); 2Department of Kinesiology and Applied Physiology, University of Delaware, Newark, DE 19713, USA; tbuckley@udel.edu (T.A.B.); cswanik@udel.edu (C.B.S.); 3Interdisciplinary Program in Biomechanics and Movement Science, University of Delaware, Newark, DE 19713, USA; 4Good Shepherd Penn Partners, Philadelphia, PA 19104, USA; 5Department of Sport, Exercise and Nutrition, School of Science and Computing, Atlantic Technological University, H91 T8NW Galway, Ireland; lisa.ryan@atu.ie (L.R.); ed.daly@atu.ie (E.D.); 6Department of Exercise Science, Thomas Jefferson University, Philadelphia, PA 19144, USA; 7Jefferson Center for Injury Research & Prevention, Thomas Jefferson University, Philadelphia, PA 19107, USA

**Keywords:** mild traumatic brain injury, collision sports, sex differences, nondisclosure, nonrecognition

## Abstract

This study examined the prevalence of intentionally unreported and potentially unrecognized concussions in community rugby players and whether nondisclosure reasons vary by sex, position, or playing history. An online survey was completed by 1037 players (41.0% female; mean age 31.6 ± 11.3 years; 10.1 ± 8.1 years playing) who reported diagnosed, unreported, and unrecognized concussions. Poisson regression models estimated prevalence ratios (PRs), and Fisher’s exact tests compared reasons for nondisclosure. The diagnosed, unreported, and unrecognized concussion rates were 66.5%, 32.4%, and 42.2%, respectively. Players with diagnosed concussions had a 7.2-fold higher prevalence of nondisclosure and a 2.3-fold higher prevalence of nonrecognition. A longer playing history was linked to greater nondisclosure (PR: 1.2), and males had a higher prevalence of nonrecognition (PR: 1.4). Position and sex were not associated with nondisclosure; position and playing history did not affect recognition. While nondisclosure reasons were mostly consistent across demographics, players with a history of concussion were more likely to report avoiding removal from games or practices (38.5% vs. 13.6%, *p* = 0.021). Concussions are common in community rugby, with high rates of underreporting and unawareness, influenced by experience and prior concussions. These findings underscore the need for better education and reporting systems to improve player safety.

## 1. Introduction

The global appeal of rugby union, henceforth referred to as ‘rugby’, has expanded rapidly, fueled by increased participation, its 2016 reintroduction to the Olympics, and the development of new professional leagues for men and women [1]. Rugby is a collision sport involving frequent, high-intensity collisions such as tackles and rucks, contributing to its high concussion risk [2]. With a rate of 28.3 concussions per 10,000 athlete exposures (AEs), rugby has the highest concussion incidence among all contact sports; however, rates may vary by level of play, sex, and player position (~19.08 to 37.42 concussions per 10,000 AEs) [3]. These variations in concussion rates highlight the need for accurate concussion assessment and reporting across playing levels and demographics. Nondisclosure of concussions is common in sport and is influenced by demographic factors, including sex, playing position, and experience [4]. Male athletes are more likely than females to conceal a potential concussion, and differences in nondisclosure depending on position and playing history have been observed among elite rugby players [4,5]. However, these patterns remain unexplored in community rugby players. Beyond demographics, internal and external pressures, such as game context (practice, championship game, etc.) or influences from coaches and parents, also contribute to motivations for nondisclosure [6,7]. Despite extensive research on concussion nondisclosure in athletes [4,7,8], these motivations and contributing factors remain unexplored in community rugby, where access to regular sports medicine providers is often lacking [9]. Furthermore, globally, community rugby players, unlike referees and coaches, are not required by World Rugby to complete a course of pre-participation concussion education, increasing the risk of potentially unrecognized concussions. Barriers to recognition include inconsistent concussion definitions, limited access to qualified medical personnel for player assessment, particularly at the amateur level, and prevailing cultural attitudes towards playing through injury [10]. Collectively, these factors may all contribute to potential concussion nonrecognition in community rugby.

Community-level players make up the majority (>60%) of registered participants, and female participation is rising [1,11]. Yet, research remains largely focused on elite and professional male players [9,12,13]. In community rugby, understanding concussion incidence in relation to history of diagnosis, playing experience, and position could help pinpoint areas where interventions are most needed. Given the long-term consequences of concussion and the risks associated with nondisclosure and nonrecognition, it is critical to assess its prevalence among community rugby players [9,14,15]. Therefore, the primary purpose of this study was to identify the prevalence of potentially unrecognized and intentionally unreported concussions in community rugby players, as interventions to address these types will likely need to differ. A secondary purpose was to explore how motivations for nondisclosure differ by sex, position, and playing experience, given the known differences in concussion rates across these variables. We hypothesized that community rugby players would report a history of undiagnosed and potentially unrecognized concussion. Moreover, we hypothesized that nondisclosure rates would be higher among males, forwards, and those with longer playing histories compared to females, backward-position players, and those with shorter histories.

## 2. Materials and Methods

### 2.1. Participants

Current and former community rugby players were recruited from March through April 2020 via rugby specific forums on social media platforms to complete an online survey detailing their rugby history, sports participation, and injury/concussion history. Eligibility criteria included self-reporting, being 18 years or older, and having played at least one year of tackle (full contact) rugby. Participants provided informed consent via the Qualtrics survey platform (Qualtrics, 73 Provo, UT, USA) as approved by the host’s institutional review board.

### 2.2. Procedures

Online recruitment efforts yielded 1376 survey responses. Of these, 1054 individuals completed all survey items. Following eligibility screening, 1037 participants were retained for analysis, representing a completion rate of approximately 75.4% [9]. These participants completed a reliable (ICC: 0.92) injury history questionnaire to identify their lifetime histories of diagnosed, potentially unrecognized, and intentionally unreported concussions [16,17] (Appendix A). Diagnosed concussions were self-reported (yes; no) in response to the question “Have you ever suffered a concussion?” Consistent with previous studies using this questionnaire, no formal definition of concussion was provided [16,17]. Potentially unrecognized concussions were coded as “yes” if participants affirmed experiencing signs/symptoms of concussion and if the number of occurrences exceeded those reported as diagnosed concussions. This operational definition is consistent with prior studies using the same validated questionnaire [16,17]. While no formal clinical definition was provided, this approach allows for the identification of symptom-based events that may reflect undiagnosed concussions, acknowledging the inherent limitations of retrospective self-report. Lastly, unreported concussions were coded as “yes” if participants affirmed suffering a concussion and not reporting it; a follow up questioned the reasons for nondisclosure (Appendix A).

### 2.3. Statistical Analysis

Demographics were summarized as means and standard deviations or frequencies and proportions where appropriate. Concussion nondisclosure and nonrecognition rates were reported as overall frequencies. Poisson regression models with robust standard errors were used to estimate prevalence ratios (PRs), with 95% confidence intervals (CIs) being used for intentionally unreported and potentially unrecognized concussions. This approach is recommended for binary outcomes with non-rare events, as it provides consistent estimates of PRs and accounts for model misspecification [18]. All independent variables and their categorizations were defined *a priori*, before examining any outcome data, and selected based on the existing literature [4,19,20]. Multivariable models included all independent variables of interest: diagnosed concussion history (yes; no), sex (male; female), position (forward; back), and years of rugby played (<8 years; >8 years). Years played was dichotomized based on the sample’s median playing history of 8 years. Model fit was assessed using the Pearson goodness-of-fit statistic, which showed no evidence of overdispersion, confirming the appropriateness of the Poisson model. Multicollinearity was evaluated using variance inflation factors (all < 2.5), indicating no multicollinearity among predictors. Fisher’s exact test was used to compare the distributions of nondisclosure reasons by diagnosed concussion history (yes/no), sex, player position, and years of rugby played. All analyses were performed with Stata SE (Version 18.5, StataCorp, College Station, TX, USA).

## 3. Results

The study included 1037 community-level rugby players (59% male, mean age: 31.6 ± 11.3 years). The participants averaged 10.1 years of rugby experience, with roughly two-thirds self-reporting a diagnosed concussion (Table 1). There were no clinically meaningful differences in demographics (i.e., age and sex) between individuals with and without a history of diagnosed concussion, as noted by the very small effect sizes; however, those with a history of concussion had longer playing histories and a higher prevalence of unreported and unrecognized concussions (Appendix A). Among the 336 participants with an unreported concussion, 314 (93.5%) had a history of diagnosed concussion. Those with a history of intentionally unreported concussion had longer playing careers (mean difference: 1.9 years), a greater prevalence (mean difference: 39.9%) and number of diagnosed concussions (mean difference: 2.7), and a higher prevalence of unrecognized concussion (mean difference: 33.5%) (Appendix A).

Among this sample, 32% intentionally did not report a concussion and 42% potentially did not recognize a concussion. Adjusted models showed that individuals with a history of diagnosed concussion had a prevalence of concussion nondisclosure that was 7.2 times higher (95% CI: 4.6, 10.6) compared to those without a history of diagnosed concussion. Those with longer playing histories had a 20% higher prevalence (PR: 1.2 [95% CI: 1.0, 1.4]) of concussion nondisclosure compared to those with shorter playing histories. There was no significant association between position or sex and nondisclosure (Table 2).

The adjusted models revealed that individuals with a history of diagnosed concussion had a 2.3-fold higher prevalence (95% CI: 1.9, 2.9) of concussion nonrecognition compared to those without a history of diagnosed concussion. Males had a 39% higher prevalence (PR: 1.39 [95%CI: 1.2, 1.6]) of unrecognized concussion compared to females. There were no significant associations between position or playing career length and concussion nonrecognition (Table 3). Among those who intentionally did not report a concussion, self-reported reasons for intentional nondisclosure such as “Did not think it was serious” and “Did not think it was a concussion” did not vary between males and females (*p*’s > 0.353), forwards and backs (*p*’s > 0.311), or those with different lengths of playing history (*p*’s > 0.107). Individuals with/without a history of diagnosed concussion did not differ in terms of motivation for nondisclosure across all but one outcome (*p*’s > 0.076), whereby individuals with a history of concussion reported a higher prevalence of not wanting to be pulled from future games/practices compared to those without a history of diagnosed concussion (38.5% and 13.5% respectively, *p* = 0.021) (Table 4).

## 4. Discussion

The primary finding of this large study on community-level rugby players is that in addition to a high prevalence of diagnosed concussion history (66.5%), a meaningful proportion of players also noted intentionally not reporting (32.4%) and/or potentially not recognizing a concussion (42.2%). A secondary finding is that players with a longer playing history and previous concussion diagnoses were more likely to have intentionally not reported a concussion, with no differences according to sex or position. Lastly, significant motivations for nondisclosure were centered around “not missing future games or practices”, especially in players with a previous diagnosis of concussion. These findings suggest a high prevalence of unrecognized and nondisclosed concussions in community rugby players, underscoring the need for targeted education and regular sports medicine access.

The prevalence of nondisclosure and unrecognized concussions in our study is considerably higher than previous research that encompasses a broad range of ages, ability levels, and sports [7,17,19,20]. The elevated rates in our sample may reflect several factors. First, our cohort’s age likely increases exposure to concussions and opportunities for unreported or unrecognized concussion, particularly as rugby is one of the highest-risk sports for concussion and has not been included in studies examining collegiate athletes [17,20] or high school athletes [19]. Second, collegiate student athletes typically have access to sports medicine professionals at games and practices, paired with mandatory concussion education, which likely improved their recognition and reporting compared to our cohort, which typically lacks sports medicine coverage and is not required to complete a pre-season concussion module. Indeed, within our cohort, 42.4% of participants reported not having an assigned athletic trainer/physiotherapist at practice or games during their rugby career. This lack of immediate medical access suggests a potential contribution to elevated nondisclosure rates, since a concussion can only be formally identified and reported if the player seeks medical attention; thus, a lack of pitch-side care would delay and possibly deter injury identification and reporting.

The nondisclosure rates in our sample are also greater than those reported in other community rugby cohorts. A UK study found that 35% of adult players had at least one concussion in the previous two seasons, and 43% did not report concussion symptoms—a rate twice as high as elite-level rugby players (17–20%) [21]. In a U.S. cohort, 61.9% of respondents reported a history of concussion, with 27.7% noting at least one concussion that was not formally reported [22]. However, the aforementioned study excluded players with more than 10 years of playing experience, which may cause rates to be underestimated, as our findings show that players with longer playing histories were more likely to both have a history of concussion and fail to disclose a concussion. In summary, the increased prevalence of nondisclosed and unrecognized concussions in the community-level players analyzed here may be influenced by the older age of our study participants, the heightened concussion risks in rugby, and the participants’ limited access to medical professionals.

A secondary purpose was to identify predictors of nondisclosure and nonreporting. Concussion history was strongly associated with nondisclosure, with the prevalence of nonreporting being seven-fold higher among those with a history of concussion. This aligns with findings in elite players, where concussion history strongly predicts nondisclosure [5]. Tadmor et al. reported that elite rugby league players with 1–2 concussions in the past two seasons were twice as likely to not report a concussion, and those with 3–5 concussions were more than five times likelier to refrain from reporting a concussion. This pattern was also found in both male and female players in the US, as well as in collegiate non-rugby athletes [7,22]. The link between concussion history and concussion nondisclosure may be explained by well-supported findings suggesting that the knowledge and attitudes of elite players do not predict reporting, which is more often influenced by external factors, including game situations and possible substitutions from play [23]. While previous studies have shown differences in concussion nondisclosure rates depending on sex and position [24], there was no significant difference in our sample. It is possible that choosing to report symptoms is a highly personalized and subjective decision based on playing experience rather than sex and position, based on evidence from elite rugby players [12,25]. Although a longer playing history increases the risk of concussion, it was not a significant predictor of nondisclosure in our sample, suggesting that concussion history and longer playing history may play a larger role in reporting behavior.

Herein, having a diagnosed concussion history more than doubled the prevalence of not recognizing a concussion. It is important to note that injury dates were not collected, so the order of diagnosed and unrecognized concussions is unable to be determined; as such, it is plausible that an athlete had an unrecognized concussion followed by a diagnosed concussion (or vice versa). This finding is consistent with a study of high school athletes wherein nondisclosure was linked to limited symptom recognition [26]. However, concussion history did not influence concussion knowledge in these high school athletes, suggesting that exposure may not align with behavior [26]. This principle has also been demonstrated in adult female rugby players in the UK, where even with exposure to specific concussion awareness training (HEADCASE), players were not more likely to report concussive symptoms [25]. Herein, males had a greater prevalence of not recognizing a concussion compared to females. This finding may be explained by previous research suggesting that female athletes are more likely to experience increased symptoms such as migraine headaches and be more honest about reporting these symptoms [27]. Although a longer playing history was linked with a higher prevalence of not recognizing a concussion, this was not significant when adjusting for other variables. While research on the relationship between playing history and concussion recognition is limited, players with longer careers may be more susceptible to recall bias, though this influence appears less significant in our sample than factors like concussion history, which play a larger role in recognition [22]. Despite differences in concussion incidence by player position in elite athletes [24], no differences in terms of recognition were observed between forwards and backs, suggesting that position may influence recognition less than concussion history. While not examined in this study, this may reflect team-wide dissemination of educational programs, though this is not a requirement for participation, rather than position-specific training [28,29]. Additionally, trained medical professionals may be able to recognize concussions more effectively than community players who must self-recognize in the absence of medical support. These findings also align with ongoing international efforts to improve concussion recognition and reporting [5,21,22,30,31,32].

Significant motivations for nondisclosure in our study primarily revolved around avoiding missed practices or games, differing from other studies that have identified factors such as perceived seriousness and obsessive passion as key predictors of nondisclosure in community players [33], with additional concerns of self-management of concussions in addition to lack of knowledge and poor communication with internal and external team support observed in women’s elite and sub-elite players [34]. However, avoiding future missed practices or games was the sole significant motivator behind nondisclosure in a study of former NCAA athletes [4]. Herein, there is likely no single factor that explains nondisclosure. Instead, educational gaps and external environmental pressures appear to play a role that was not explored herein, highlighting the need for systemic reporting programs and further qualitative analysis of unreported concussions [35].

The limitations of this study include potential sampling bias, with healthier players or those with a concussion history more likely to participate, possibly over-representing certain groups [9]. Social desirability bias may have led some participants to underreport concussions, affecting prevalence data [17]. As noted previously, injury dates were not collected, so the order between diagnosed and unrecognized concussions is unknown, limiting interpretations to correlations only. Additionally, given the increased awareness and stricter guidelines since the late 2000s, participants may have sustained injuries during a time when concussion knowledge was more limited, contributing to nonrecognition [36]. Although novel, these findings are limited to predominantly American community rugby players, restricting generalizability to other sports or populations. Notably, our sample included a higher proportion of female players than globally estimated (25% vs. 41% herein), which may reflect oversampling due to our recruitment strategy [37]. To improve representativeness in future research, partnering directly with rugby clubs or regional rugby organizations may facilitate broader recruitment and reduce sampling bias. Such collaborations could enable more systematic data collection, improve access to current players, and allow for stratification by playing level, geography, or club resources. Moreover, participants reported lifetime concussion histories without exact injury dates; therefore, recall bias, injury order (e.g., undiagnosed concussion preceding a diagnosed concussion), and participant honesty is a concern. However, this injury history survey is well utilized and validated [9,16,17]. Importantly, the construct of “potentially unrecognized concussion” was operationalized based on symptom endorsement exceeding the number of diagnosed concussions, consistent with prior studies [16,17]. While this approach allows for potential insight into underrecognized injuries, it is inherently limited by self-report and lacks clinical confirmation; future research should consider prospective designs or clinical validation to refine this construct and explore how alternative definitions may influence prevalence estimates. Additionally, there may be other factors not explored herein that contribute to reporting, such as education level, access to direct medical care, timely care, team culture, temporal relationship between events, number of unrecognized/unreported concussions, etc. Importantly, we did not have the ability to separate analyses by current versus former rugby player status, so differences in reporting may occur depending on generation and playing status (i.e., current versus former), and this should be considered in future studies. Finally, this study did not explore behaviors or attitudes before or after injury, limiting insights into how prior concussions may influence current views or actions.

## 5. Conclusions

In this cohort of community rugby players, we observed a high prevalence of concussion history, with considerable (or high or concerning) rates of intentionally unreported and potentially unrecognized concussions. Key predictors included a history of diagnosed concussion and longer playing careers. These findings highlight the need for greater awareness among coaches, players, and support staff about the prevalence of concussions. Broad, systematic education and reporting systems are critical for enhancing player safety and welfare at all levels and should be considered in future research. To improve detection and prevention, community rugby teams should consider implementing standardized sideline assessment protocols, increasing access to trained medical personnel during matches, and promoting a culture of safety that encourages symptom reporting without stigma. Educational initiatives should emphasize the signs of concussion and the long-term risks of underreporting, particularly for players with extensive playing histories or prior concussions.

## Figures and Tables

**Table 1 sports-13-00278-t001:** Participant demographics.

	Overall (*n* = 1037)	Male (*n* = 612, 59%)	Female (*n* = 425, 41%)
Age (y), Mean (SD)	31.6 (11.3)	33.7 (12.8)	28.7 (7.8)
Position, *n* (%)			
Back	406 (39%)	241 (39%)	165 (39%)
Forward	631 (61%)	371 (61%)	260 (61%)
Years of Rugby Played, Mean (SD)	10.1 (8.1)	12.2 (9.0)	7.1 (5.2)
History of Diagnosed Concussion, *n* (%)	690 (67%)	412 (67%)	278 (65%)
Number of Diagnosed Concussion, *n* (%)	2.0 (2.6)	2.2 (2.7)	1.8 (2.4)
Intentionally Unreported Concussion, *n* (%)	336 (32%)	207 (34%)	129 (30%)
Potentially Unrecognized Concussion, *n* (%)	438 (42%)	294 (48%)	144 (34%)

**Table 2 sports-13-00278-t002:** Prevalence ratios of not disclosing concussions.

Independent Variable	% (*n*) of Nondisclosure (*n* = 336)	Univariate (95% CI)	Multivariate (95% CI)
Diagnosed Concussion			
Yes	93.5% (314)	**7.18 (4.75, 10.85)**	**7.02 (4.64, 10.61)**
No	6.5% (22)	1.0 (Ref)	1.0 (Ref)
Position			
Back	37.8% (127)	0.94 (0.79, 1.13)	0.97 (0.82, 1.15)
Forward	62.2% (209)	1.0 (Ref)	1.0 (Ref)
Sex			
Male	61.6% (207)	1.11 (0.93, 1.33)	1.03 (0.87, 1.23)
Female	38.4% (129)	1.0 (Ref)	1.0 (Ref)
Years Playing Rugby			
>8 years	59.8% (201)	**1.37 (1.14, 1.64)**	**1.19 (1.00, 1.42)**
<8 years	40.2% (135)	1.0 (Ref)	1.0 (Ref)

Note: Model 1 (univariate) estimates the prevalence ratio (PR) of nondisclosure based on the independent variable only. Model 2 (multivariate) adjusts for sex, player position, and years of rugby played, where applicable. Bolded PRs indicate statistically significant associations. CI = confidence interval.

**Table 3 sports-13-00278-t003:** Prevalence ratios of not recognizing concussions.

	% (*n*) of Nondisclosure (*n* = 336)	Univariate (95% CI)	Multivariate (95% CI)
Diagnosed Concussion			
Yes	361 (82.4%)	**2.36 (1.91, 2.91)**	**2.34 (1.90, 2.88)**
No	77 (17.6%)	1.0 (Ref)	1.0 (Ref)
Position			
Back	177 (40.4%)	1.95 (0.91, 1.22)	1.06 (0.93, 1.22)
Forward	262 (59.6%)	1.0 (Ref)	1.0 (Ref)
Sex			
Male	294 (67.1%)	**1.42 (1.21, 1.66)**	**1.39 (1.19, 1.63)**
Female	144 (32.9%)	1.0 (Ref)	1.0 (Ref)
Years Playing Rugby			
>8 years	247 (56.4%)	**1.19 (1.03, 1.38)**	1.01 (0.87, 1.17)
<8 years	191 (43.6%)	1.0 (Ref)	1.0 (Ref)

Note: Model 1 (univariate) estimates the prevalence ratio (PR) of nondisclosure based on the independent variable only. Model 2 (multivariate) adjusts for sex, player position, and years of rugby played, where applicable. Bolded PRs indicate statistically significant associations. CI = confidence interval.

**Table 4 sports-13-00278-t004:** Motivation for intentional concussion nondisclosure (*n* = 336).

	Did Not Think It Was Serious	Did Not Know It Was a Concussion	Did Not Want to Be Pulled from Current Game/Practice	Did Not Want to Be Pulled from Future Games/Practices	Did Not Want to Let Teammates Down	Would Have if It Was a Less Important Game/Practice	Other
Responded ‘Yes’	218 (64.9%)	181 (53.9%)	158 (47.0%)	124 (36.9%)	110 (32.7%)	35 (10.4%)	22 (6.5%)
Male	134 (64.7%)	116 (56.0%)	95 (45.9%)	72 (34.8%)	71 (34.3%)	28 (13.5%)	12 (48.0%)
Female	84 (65.2%)	65 (50.4%)	63 (48.8%)	52 (40.3%)	39 (30.2%)	16 (12.4%)	12 (48.0%)
*p*-value	1.000	0.368	0.653	0.353	0.475	0.868	0.393
Backs	84 (66.1%)	69 (54.3%)	55 (43.3%)	42 (33.1%)	42 (33.1%)	18 (14.2%)	8 (6.3%)
Forwards	134 (64.1%)	112 (53.6%)	103 (49.3%)	82 (39.2%)	68 (32.5%)	26 (12.4%)	17 (8.1%)
*p*-value	0.725	0.911	0.311	0.294	1.000	0.739	0.669
Diagnosed Concussion History	203 (64.7%)	167 (53.2%)	152 (48.4%)	121 (38.5%)	106 (33.8%)	44 (14.0%)	25 (8.0%)
No Diagnosed Concussion History	15 (68.2%)	14 (63.6%)	6 (27.3%)	3 (13.6%)	4 (18.2%)	0 (0.0%)	0 (0.0%)
*p*-value	0.821	0.383	0.076	0.021 *	0.162	0.093	0.390
Years Playing Rugby <8	133 (66.2%)	108 (53.7%)	89 (44.3%)	67 (33.3%)	70 (34.8%)	27 (13.4%)	15 (7.5%)
Years Playing Rugby ≥8	85 (63.0%)	73 (54.1%)	69 (51.1%)	57 (42.2%)	40 (29.6%)	17 (12.6%)	10 (7.4%)
*p*-value	0.562	1.00	0.223	0.107	0.344	0.870	1.00

Note: The frequencies of male/female, backs/forwards, and diagnosed concussion (yes/no) are based upon those who stated that they did not disclose a concussion (*n* = 336). The participants could select multiple reasons for nondisclosure. “Other” includes free-text responses entered into an open-ended field. *: significance at *p* < 0.05.

## Data Availability

The data presented in this study are available on request from the corresponding author.

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
