# Peer review of "Unrecognized and Unreported Concussions Among Community Rugby Players"

_sports, 2025, doi:10.3390/sports13080278_

Round 1

Reviewer 1 Report

Comments and Suggestions for Authors

This is a well-written paper on a relevant and timely topic. The sample size is sufficient. Procedures and statistical analyses seem appropriate.

The primary shortcoming lies in the attempt to determine “unrecognized concussion”. – The self-reported symptoms are very likely not a fully dependable measure to assess this quantity. Then again, I have no better suggestion for how to determine this variable and I agree that it is a worthwhile variable to investigate.

Overall, I have only minor comments and support its publication.

Minor comments:

Throughout the manuscript: references to papers, tables, or figures are placed after the full stop. This is confusing, as it separates the reference from the relevant sentence. I recommend placing the citation brackets before the period to ensure clarity and to maintain the logical flow of the text.

Line 121: „self110 reporting“

Line 120: “+ 109 11.3 years”

Line 126,

line 127: Again, there are undeleted line numbers in the text, that should have been cleaned up!

Line 265: Start a new paragraph for the limitations?

Reviewer 2 Report

Comments and Suggestions for Authors

I commend the authors on completing this important work. The authors do an excellent job of laying out the scope of the problem with concussion in community-level rugby players in their introduction. They further demonstrated that most rugby players are amateurs and have not been well studied. 

I found the study to be novel and provided useful information on why concussion is being underrecognized and underreported in this cohort. These findings can be applied more broadly to other collision and contact sports. 

They have addressed two populations that merit more attention when discussing concussion, Rugby players and amateur players. Rugby is a collision sport with high risk for concussion. There tends to be more focus on professional or collegiate players and not as much focus on community players. This study shows why it is important to study amateur sports. There are far more people engaged in amateur sports than in professional sports and so this recognition of concussion in this cohort is extremely important. 

The article was well written, and the data and analyses were appropriately represented. This paper will primarily be of interest to medical providers caring for rugby players, but providers for players in other contact and collision sports. 

The authors indicate that female participation is rising, but did not provide data on what percentage of rugby players are female. This is useful because it will help us to see if women were either over or undersampled in this cohort. According to world rugby (https://www.world.rugby/organisation/about-us/womens), “more than 25% of current players are women.” If women were oversampled in this study, was there something about the methodology and how subjects were recruited that may have contributed to this discrepancy. Is it possible that this oversampling of women impacted the results?

The scientific community will benefit from having this paper available as a reference.

 There are some areas of the paper that require grammar editing, but I found the overall content to be well organized and easy to understand. The conclusion could certainly be more robust with recommendations on concussion prevention and how to improve detection in these community rugby players. 

Reviewer 3 Report

Comments and Suggestions for Authors

Thank you for your submission on concussion reporting in community rugby. Although the topic is relevant, I have several concerns that need to be addressed before this manuscript can be considered for publication.

Major Issues

1. Sample Representativeness Concerns

While social media recruitment is increasingly common in health research, you need to provide more detailed discussion of potential selection bias. Players willing to complete online surveys about concussions may differ systematically from the general rugby population. I would recommend including demographic comparisons with established rugby population data (e.g., from national rugby organizations) to help readers assess generalizability. Additionally, the response rate and recruitment details should be reported more transparently.

2. Statistical Approach Needs Improvement

The multivariate analysis lacks sufficient detail. Which variables were included in the final model? How did you handle collinearity between age and playing experience? Your odds ratios are presented without confidence intervals in several places, and there is no discussion of model fit or assumption checking. The categorization of continuous variables (e.g., years playing) appears arbitrary - where is the justification for this?

3. Construct Definition and Measurement

Your definition of "potentially unrecognized concussion" requires better justification. While I understand that measuring unrecognized events presents inherent challenges, you should provide more detailed rationale for your operational definition and discuss how it relates to clinical definitions. Consider including sensitivity analyses or discussing how different definitions might affect your results. The construct appears reasonable for exploratory research, but limitations should be acknowledged more explicitly.

4. Literature Review Needs Strengthening

You have missed several key papers in this field, particularly recent epidemiological studies. More importantly, your discussion of concussion management protocols does not reflect the latest Amsterdam Consensus Statement (2023), which provides updated evidence-based recommendations for sport-related concussion management. This raises questions about your familiarity with the most recent developments in the field.

Minor but Important Issues

Sample size calculation is missing. How do you know you have adequate power for your analyses?

The response rate is not reported - what percentage of those who saw your survey actually completed it?

Multiple comparisons are not properly accounted for, which inflates Type I error risk

Figure 2 is poorly labeled and difficult to interpret

Several grammatical errors suggest insufficient proofreading

Specific Technical Concerns

Line 156: You state "significant association" but p=0.067 - this does not meet conventional significance levels

Table 3: Why are some categories combined post-hoc? This suggests possible data dredging

The discussion overstates findings - "demonstrates" should be changed to "suggests" throughout

Statistical methods section needs more detail about handling of missing data

Recommendation

While the research addresses an important question using appropriate methods for this type of exploratory study, several areas need strengthening to enhance the manuscript's contribution. The issues raised are addressable through revision rather than fundamental redesign.

I would suggest the authors consider:

Partnering with rugby clubs for more representative sampling

Using more rigorous statistical approaches with expert consultation

Conducting proper power analysis

Strengthening the validity of key outcome measures

The manuscript would benefit from substantial revision, preferably with input from someone with stronger epidemiological expertise.

Round 2

Reviewer 3 Report

Comments and Suggestions for Authors

To the Authors:

Thank you for your revised manuscript. And for your detailed response. I have read them both carefully. I appreciate your effort.

I am satisfied with your revisions.

Sample Representativeness: You added a good discussion about selection bias. You explained the limitations of available data. This is good. Future research plans are good.

Statistical Approach: Your explanation about the multivariate analysis is clearer. VIF calculation is good. You confirmed all results have confidence intervals. This is important.

Construct Definition: Your definition of "unrecognized concussion" is clearer now. You explained the limitations. This is good.

Literature Review: You added the 2023 Amsterdam Consensus Statement. This shows your expertise. Thank you for this update.

Other Issues: You fixed all other issues. Like sample size and response rate. And grammatical errors.

Your revisions improved the manuscript a lot. I think it is ready for publication.

Sincerely,